# Constant Tension Control System of High-Voltage Coil Winding Machine Based on Smith Predictor-Optimized Active Disturbance Rejection Control Algorithm

**DOI:** 10.3390/s23010140

**Published:** 2022-12-23

**Authors:** Yuming Ai, Baocheng Yu, Yanduo Zhang, Wenxia Xu, Tao Lu

**Affiliations:** 1School of Computer Science and Engineering Artificial Intelligence, Wuhan Institute of Technology, Wuhan 430205, China; 2School of Literature and Media, Hubei University of Arts and Science, Xiangyang 441053, China

**Keywords:** high-voltage coil, constant tension, Smith predictor, auto disturbance rejection control

## Abstract

In the production process of high-voltage coils, a constant tension control system is designed to improve the quality of the transformer. The system is composed of a controller, execution structure, detection structure, etc. The active disturbance rejection control (ADRC), optimized by the Smith predictor (SP), is adopted to achieve constant tension control. The experiment results show that the tension control system based on the SP-ADRC has higher control accuracy, shorter stabilization time and stronger anti-interference ability compared with the traditional PID algorithm. The actual experiment shows that the constant tension control system of the high-voltage coil winding machine based on SP-ADRC has a superior control effect and high practical value.

## 1. Introduction

In recent years, with the development of science and technology, the automatic winding of high-voltage transformer coils has been realized. In the process of automatic winding, the metal wire of the transformer high-voltage coil has a certain elastic coefficient [1,2]. The acceleration of wire conveying speed and the change in reel radius will change the winding tension. Improper tension control will seriously affect the winding quality of the transformer winding machine [3,4,5,6]. When the tension is too small, it will lead to the relaxation, accumulation, collapse, etc., of the conductor or insulating tape. When the tension is too large, it will lead to deformation, excessive tension or even fracture of the conductor. Therefore, in the production process of high-voltage coils, the performance of tension control directly affects the quality of the transformer, which is the key technology of the transformer winding machine equipment control [7,8,9,10,11,12,13].

Recently, several tension-control systems were designed for winding tension and increasing the precision of the workpiece. Li et al. [14] developed a wire tension control system based on closed-loop PID control. They built a mathematical model of a wire tension control system composed of a DC servo motor, linear motion platform and tension sensor. Then, the optimal PID control parameters were calculated by Matlab and Simulink Toolbox. The adaptive control can be realized in the tension control system, where 5–15 N is the optimized wire tension with an error of 5%. Chen et al. [15] developed a high-precision constant wire tension control system, which was based on the structural improvement and PID closed-loop controller. This constant wire tension control system has high control accuracy, quick response and strong anti-interference capability. Hong et al. [16] presented a tension-control system based on the control algorithm ESO-based ASMC. They demonstrated that by estimating and compensating for the total disturbance using the extended state observer, the influence of the total disturbance on the control accuracy can be reduced, and the dynamic performance of the system can be improved. Fang et al. [17] presented a control strategy for smooth wire sending and constant wire tension in a multi-wire saw machine. They divided the wire-sending state into several sections and designed a novel feedforward multi-conditioned position controller to achieve good control performance. Wei et al. [18] designed a constant tension and constant linear speed winding system based on a SIEMENS S120 driver. Their design applied SIEMENS PROFINET and Drive-CLiQ field-bus to a tension control winding system, which effectively improved production efficiency and process performance. He et al. [19] studied the method of fuzzy adaptive control to achieve the inhibition of tension vibration for the winding tension control system. They found that the tension vibration of the fuzzy adaptive control mode was more effective than the PID control mode. Huang et al. [20] developed a tension-control system for transformer insulation layer winding. They designed a variable universe fuzzy PI controller to adjust the unwinding speed. Compared with the traditional PID control method, this method can reduce the influence of tension disturbance in the insulation winding process of the elliptical iron core transformer. However, the tension changes within the range of 4%, which needs to be improved.

The time delay will introduce additional phase lag and reduce the phase margin, which is the main limitation of systems that have a time delay. Now, a lot of Smith predictor (SP)-based methods [21] have been developed for time-delayed systems. The main advantage of SP-based methods is that the delay can be removed, which can improve the closed-loop bandwidth.

With the development of ADRC, researchers extended ADRC applications to time-delayed systems. The basic reason why ADRC has limitations is that the two signals (u and y) entering ESO are not synchronized due to time delay, which leads to inaccurate estimation of the “generalized interference” and reduces the control performance [22]. Therefore, in order to eliminate the influence of time delay in ADRC design and improve stability, some modifications were proposed. The Smith predictor was used to output signals in advance to achieve synchronization, which is expressed as SP-based ADRC (SP-ADRC) [23].

Ba et al. [24] designed a drill bit stick–slip vibration and active disturbance rejection controller. The results show that the designed Smith predictor-based active disturbance rejection controller has better control performance, which can effectively suppress the sudden change in torque and speed.

Chen et al. [25] introduced the improved SP-ADRC of the time delay system and simulates the boiler turbine. The results show that SP-ADRC can handle any large time delay when there is no uncertainty.

This paper creatively analyzes and establishes the mathematical model of tension in the winding process. We innovatively apply the Smith predictive optimization active disturbance rejection control algorithm to the constant tension control system of the high-voltage winding machine. Then, the performance of the control system is evaluated by a tension control experiment.

## 2. Tension Control Model of Winding System

In the winding process of the winding machine, in order to meet the requirements of the coil wire tension, there is a certain difference between the speed of the pay-off wheel and the winding wheel. Therefore, the system needs to control and adjust the speed of the winding wheel and pay-off wheel so as to ensure that the conductor has constant tension.

The tension system of the high-voltage coil winding machine mainly includes: the pay-off wheel, cylinder buffer wheel, guide wheel, tension detection wheel, wire arrangement wheel and winding wheel. Its structure is shown in Figure 1. The cylinder buffer wheel has a buffer effect when the tension changes suddenly. The guide wheel implements the guide function of the conductor. The tension detection wheel measures the conductor tension. The wire arrangement wheel moves to realize the horizontal arrangement of wires on the winding wheel. The winding wheel completes the final winding of the conductor. The conductor used in the high-voltage transformer is a 2.85 mm × 5.4 mm flat copper wire, with 9 turns and 15 layers per section.

### 2.1. Tension Detection Principle

Figure 2 shows the structure of the tension detection device. The pressure sensor is installed on the tension-detection wheel to detect the tension in the wire.

The relationship between the conductor tension *P* and pressure *F* can be described as: (1)$$P=\frac{F}{2\cos(\alpha /2)}$$

### 2.2. Mathematical Analysis of Winding System

Suppose *L* is the length of the wire between the pay-off wheel and the winding wheel, *A* is the cross-sectional area of the wire, *E* is its elastic modulus, $V_{1}$ is the linear speed of the pay-off wheel, $V_{2}$ is the linear speed of the winding wheel, *P* is the tension of the wire and $L_{0}$ is the length of the wire between the two wheels when it is not stretched. According to Hooke’s Law, the tension *P* can be defined as: (2)$$P=\frac{EA\left(L-L_{0}\right)}{L_{0}}$$

The shape variable of the wire between two wheels ΔL is calculated by: (3)$$\Delta L=L-L_{0}=\int \left(V_{2}-V_{1}\right)dt$$

Bringing it into Equation (Equation 2): (4)$$P=\frac{ES}{L_{0}}\int \left(V_{2}-V_{1}\right)dt$$
where $\omega_{1}$ is the angular speed of the pay-off wheel, $\omega_{2}$ is the angular speed of the winding wheel, $R_{1}$ is the radius of the pay-off wheel and $R_{2}$ is the radius of the winding wheel.
(5)$$\omega_{1}=\frac{V_{1}}{R_{1}},\omega_{2}=\frac{V_{2}}{R_{2}}$$
so: (6)$$P=\frac{2\pi ES}{L_{0}}\int \left(R_{2}\omega_{2}-R_{1}\omega_{1}\right)dt$$
where *h* is the thickness of the conductor, $L_{1}$ is the length of the conductor released by the pay-off shaft and $n_{1}$ is the number of turns per layer. Then we can get the change in the side area per unit time $dS_{1}$: (7)$$dS_{1}=2\pi R_{1}dR_{1}=\frac{hdL_{1}}{n_{1}}$$

Therefore, we can get:(8)$$\frac{d\omega_{1}}{dt}=\left(\frac{d\omega_{1}}{dR_{1}}\right)\left(\frac{dR_{1}}{dt}\right)=\left(\frac{d\frac{V_{1}}{R_{1}}}{dR_{1}}\right)\left(\frac{h}{2\pi R_{1}n_{1}}V_{1}\right)=-\frac{V_{1}^{2}h}{2\pi R_{1}^{3}n_{1}}=-\frac{\omega_{1}^{2}h}{2\pi R_{1}n_{1}}$$
and then: (9)$$R_{1}=-\frac{\omega_{1}^{2}h}{2\pi n_{1}}{\left(\frac{d\omega_{1}}{dt}\right)}^{-1}$$

For the same reason, we can get: (10)$$R_{2}=-\frac{\omega_{2}^{2}h}{2\pi n_{2}}{\left(\frac{d\omega_{2}}{dt}\right)}^{-1}$$
where $n_{2}$ is the number of turns per layer of the winding shaft.

Bring Equations (9) and (10) into Equation (Equation 4): (11)$$P=\frac{EAh}{L_{0}}\int \left[\frac{\omega_{1}^{3}}{n_{1}}{\left(\frac{d\omega_{1}}{dt}\right)}^{-1}-\frac{\omega_{2}^{3}}{n_{2}}{\left(\frac{d\omega_{2}}{dt}\right)}^{-1}\right]$$

During coil winding, the winding shaft speed $\omega_{2}$ can be regarded as a fixed value. In addition, the number of turns in each layer of winding shaft $n_{1}$, the number of turns in each layer of paying off shaft $n_{2}$ and the thickness of wire *h* are fixed values too. Therefore, the transfer function of the tension control model of the winding system is obtained by Laplace transform as follows.
(12)$$G\left(\;s\right)=\frac{1}{Ks+n_{1}}e^{-\tau s}$$

### 2.3. The Design of the Tension Control System

The tension control system mainly includes the controller, executive structure and detection structure. The structure of the tension control system is shown in Figure 3. The control target is the tension of the conductor; the feedback signal is sent from the tension sensor; the control signal is the set value of conductor tension; the executive mechanism of the system is the differential control of the pay-off wheel and the winding wheel.

## 3. Controller Design for SP-ADRC

The tension control system of the high-voltage coil winding machine is characterized by non-linearity, time-varying and coupling. The PID control cannot provide perfectly accurate and stable control [26]. SP-ADRC is an ADRC control algorithm optimized based on the Smith predictor [27]. The system obtains the predicted output through the Smith predictor and enters it into the ESO as the actual output parameter of the system to synchronize the two output signals, thus improving the control performance of the system. For the process control with widespread disturbance and uncertainty, under the premise that the system is stable, the disturbance suppression problem is the main objective of controller design, from which the active disturbance rejection controller is derived [28]. The extended state observer (ESO) is the core of ADRC [29]. Due to the existence of the time delay, the controller output signal U in ESO is not synchronized with the actual output signal Y of the object, thus reducing the accuracy of ESO [30,31]. The structure diagram of the SP-ADRC controller is shown in Figure 4:

The tension control system SP-ADRC can be expressed in the form of a state space as follows [32]: (13)$$\left\{\begin{array}{c} \dot{\hat{z}}=\left(A-L_{0}C\right)\hat{z}\left(t\right)+Bu\left(t\right)+L_{0}y_{p}\left(t\right)\\ u\left(t\right)=K_{0}\left(\hat{r}\left(t\right)-\hat{z}\left(t\right)\right) \end{array}\right.$$
where $y_{p}\left(t\right)$ is the delay-free output of the system, $L_{0}$ is the gain matrix of the extended state observer and $K_{0}$ is the gain matrix of the controller. According to the structure diagram of the SP-ADRC controller, we can get the Laplace transform of $y_{p}\left(t\right)$: (14)$$y_{p}\left(s\right)=y\left(s\right)+G\left(s\right)\left(1-e^{-\hat{\tau }s}\right)u\left(t\right)$$

We can see that: (15)$$K_{c}=K_{0}{\left(sI-A+L_{0}C+BK_{0}\right)}^{-1}L_{0}$$

Therefore, the transfer function from Y(s) to u(s) is: (16)$$K_{CSP}\left(s\right)={\left(1+K_{c}P_{0}\left(1-e^{-\hat{\tau }s}\right)\right)}^{-1}K_{c}$$

## 4. Results

The physical picture of the system is shown in Figure 5.

The system uses Beckhoff PLC as the controller. It collects the real-time tension of the wire through the pressure sensor and feeds it back to the controller. After running and processing by the controller algorithm, the control signal is output to the private server motor driver. Finally, the tension of the wire is stably adjusted by changing the rotation speed of the winding wheel and the pay-off wheel.

### 4.1. Simulation Analysis

In order to verify the feasibility of SP-ADRC, this paper used Simulink to design the tension control simulation experiment comparing SP-ADRC with PID. The given tension was 80 N, the speed of the pay-off wheel was 50 r/min, the speed of the winding wheel was 120 r/min, and a constant value disturbance of control quantity with amplitude of −1 V was added at 20 s. The simulation curve is shown in Figure 6.

It can be seen from the figure that when the tension system based on the PID had a larger overshoot, and, at the same time, the ability to suppress the constant disturbance was weaker. However, the tension system based on SP-ADRC had a smaller overshoot, shorter system stability time and faster disturbance suppression.

### 4.2. Constant Tension Control Test

In order to study the control effect of the system, we compared the winding system based on PID control and SP-ADRC control. The results of the tracking response and control error are shown in Figure 7 and Figure 8. The tension curve of the PID control algorithm is shown in Figure 7, and the tension curve of the SP-ADRC control algorithm is shown in Figure 8. The given tension is 80 N, the speed of the pay-off wheel is 50 r/min and the speed of the winding wheel is 120 r/min. The parameters of PID operation instructions are determined by the PID parameter-tuning method. The critical proportion method of the on-site engineering setting method is selected. The critical scaling method was as follows: (1) Setting the integral time $T_{I}$ of the regulator to the maximum, the differential time to zero and the scaling δ properly, balancing the operation for a period of time and putting the system into automatic operation; (2) the scaled δ gradually decreases, achieving a constant amplitude oscillation process and recorded the critical scale $\delta_{K}$ and critical oscillation period $T_{K}$. (3) According to the values of $\delta_{K}$ and $T_{K}$, the various parameters of the regulator were calculated. (4) Adjust the setting parameters of the regulator to the calculated values according to the operating procedure of “P first, I second, D last”. If this was not satisfactory, further adjustments could be made.

The tracking corresponds to the real-time measured-tension value of the system, and the control error is the error between the real-time tension value and the target value. It can be seen from Figure 6 that it takes 7.8 s for the tension system based on the PID algorithm to generate constant tension at the expected value. However, from Figure 7, we can see that it only takes 4.7 s for the system based on SP-ADRC to do the same. Therefore, the SP-ADRC has a faster response. The average control error of the tension system based on the PID algorithm is 1.16 N, while the average control error of the tension system based on the SP-ADRC algorithm is 0.53 N. Therefore, the SP-ADRC controller has a higher control accuracy and shorter stability time.

In order to compare the control performance of the two controllers, the maximum error, the average error and the mean square error were introduced as evaluation indicators, respectively, to compare the control performance of the two controllers. The evaluation indexes of the PID controller and SP-ADRC controller are listed in Table 1.

We can see that the maximum error and the average error of the SP-ADRC controller were smaller than the PID controller. This showed that the SP-ADRC controller had a higher control accuracy. The mean square error of the SP-ADRC controller was smaller than the PID controller, which showed that the SP-ADRC controller had better stability.

### 4.3. Analysis of Static Difference Rate

The static difference rate is an important index to evaluate the system performance. It can be expressed as follows [23]: (17)$$\delta =\frac{\Delta P}{P_{m}}\times 100\%$$
where $\Delta P=P_{\max\;}-P_{\min\;}$, $P_{\max\;}$ is the maximum tension, $P_{\min\;}$ is the minimum tension and $P_{m}$ is the average tension.

Table 2 is the analysis of the static difference rate. We can see that the static difference of the system based on PID control is 4.89%; this is larger than the system based on SP-ADRC, of which the static difference is 1.79%.

### 4.4. Disturbance Suppression Test

We conducted the disturbance suppression test by applying a sudden load to the system. The results are shown in Figure 9.

We add 10 N load when t=10 s. It can be seen in Figure 9 that the tension under the SP-ADRC control mode immediately rises to 98 N and quickly adjusts to 80 N. We can see that the tension under the PID control immediately rises to 113 N, which is 15 N more than that of the SP-ADRC algorithm. Therefore, the SP-ADRC control algorithm has a stronger anti-interference ability.

## 5. Conclusions

This paper analyzes the mathematical model of tension in the winding process of high-voltage coils and uses the SP-ADRC control algorithm to control the constant tension. We have tested the actual device, and it can be seen from the experimental results that it takes less time for the system to reach stability based on the SP-ADRC control algorithm. Further, the system based on the SP-ADRC control algorithm has higher control accuracy, smaller static difference rate and better anti-interference ability than that based on the PID algorithm.

## Figures and Tables

**Figure 1 sensors-23-00140-f001:**
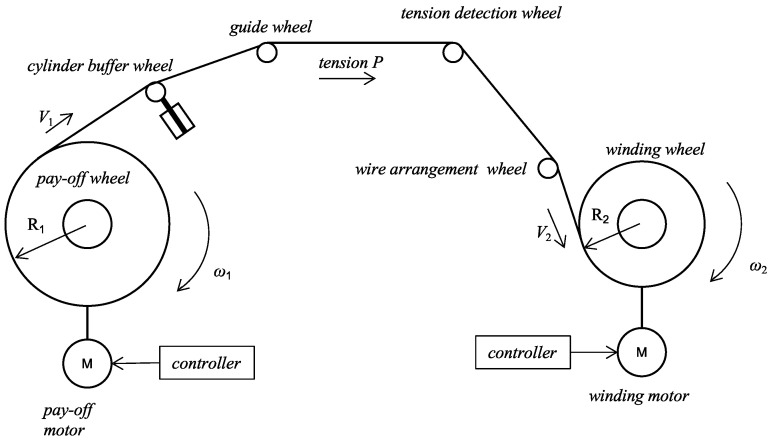
Structure diagram of the high-voltage coil winding machine.

**Figure 2 sensors-23-00140-f002:**
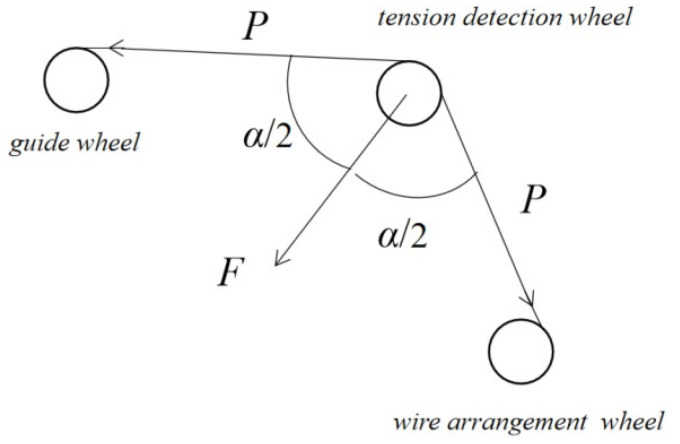
Schematic diagram of tension detection.

**Figure 3 sensors-23-00140-f003:**
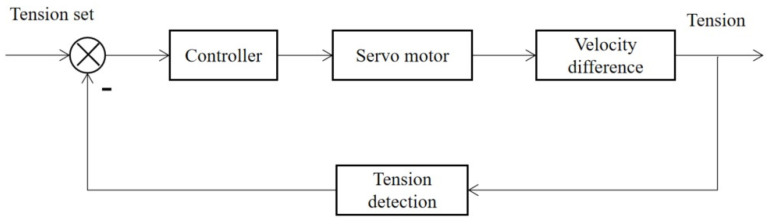
Principle block diagram of the tension control system.

**Figure 4 sensors-23-00140-f004:**
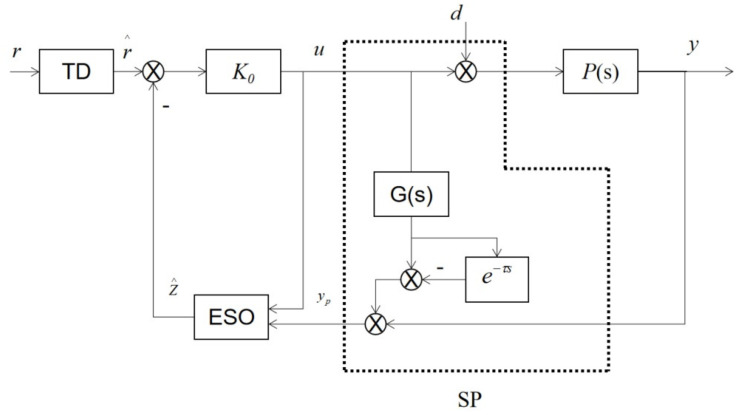
Structure diagram of the SP-ADRC controller.

**Figure 5 sensors-23-00140-f005:**
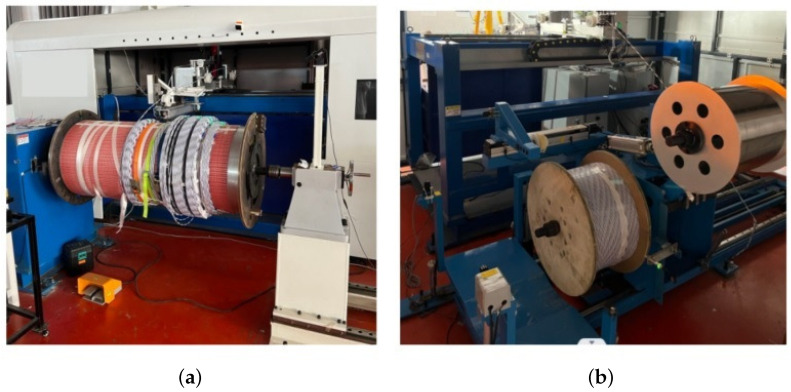
The physical picture of the winding machine; (**a**) front photo of the winding machine, (**b**) back photo of the winding machine.

**Figure 6 sensors-23-00140-f006:**
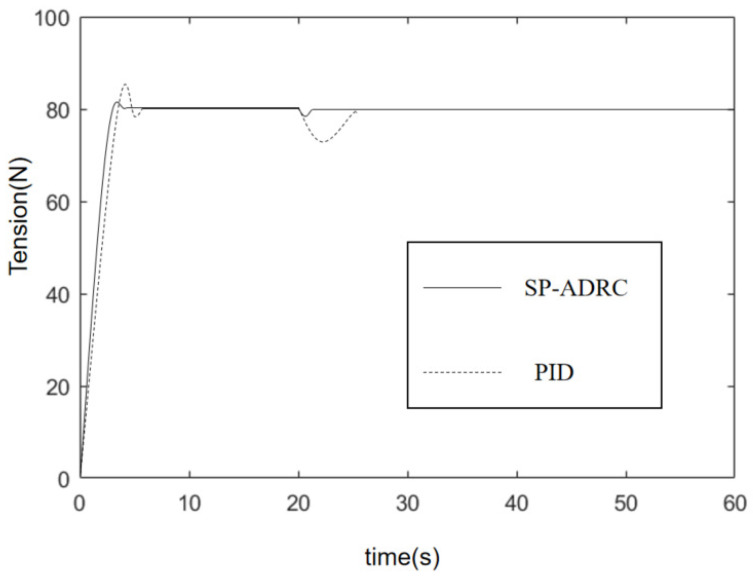
Tension simulation curve.

**Figure 7 sensors-23-00140-f007:**
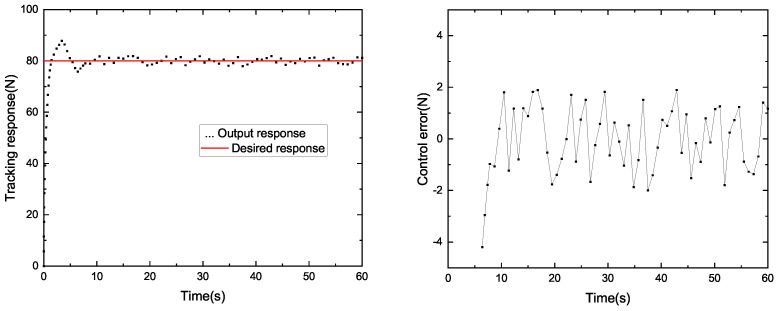
Tracking response and control error (PID).

**Figure 8 sensors-23-00140-f008:**
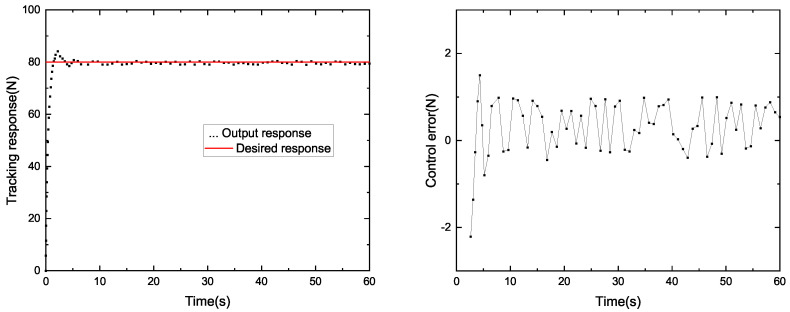
Tracking response and control error (SP-ADRC).

**Figure 9 sensors-23-00140-f009:**
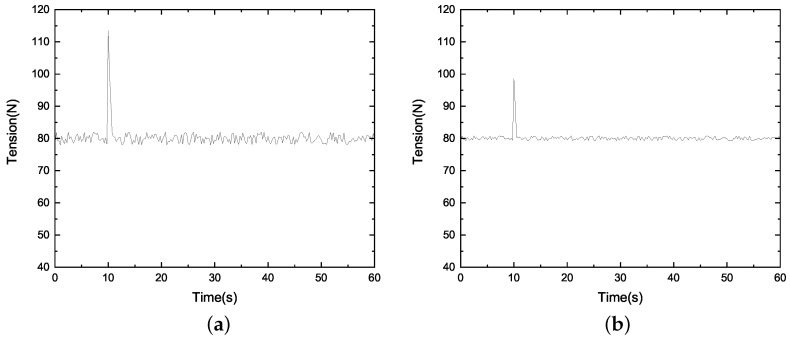
Tension local magnification curve (**a**) of the PID control and (**b**) SP-ADRC control.

**Table 1 sensors-23-00140-t001:** Control performance evaluation results.

Status	Enactment Tension/N	$P_{\max}/N$	$P_{\min}/N$
PID control	4.23	1.13	1.77
SP-ADRC control	2.21	0.57	0.48

**Table 2 sensors-23-00140-t002:** The static difference rate of tension.

Status	Enactment Tension/N	$P_{\max}/N$	$P_{\min}/N$	ΔP/N	$P_{m}/N$	δ/%
PID control	80	81.89	78.00	3.89	79.61	4.89
SP-ADRC control	80	80.44	79.01	1.43	80.10	1.79

## Data Availability

The data presented in this study are available on request from the corresponding author. The data are not publicly available due to privacy.

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
