# Peer review of "Constant Tension Control System of High-Voltage Coil Winding Machine Based on Smith Predictor-Optimized Active Disturbance Rejection Control Algorithm"

_sensors, 2022, doi:10.3390/s23010140_

Round 1
Reviewer 1 Report
· The novelty of the study should be emphasized in a more clear way.
· It is unclear why the authors have preferred to adapt auto disturbance rejection control with Smith predictor. What are the reasons behind this idea, what are the advantages over existing ones?
· How are the parameters of the control method that the authors used chosen ? This point should be clarified.
· The literature survey can be extended by adding more recent papers about the proposed subject. With this way, the introduction section can be improved.
· All the equations taken from the literature should be cited.
· Transfer function of the Tension Control Model of Winding System should also be mentioned in the study. Its order, being linear or not should be clearly mentioned.
· Figure 3 does not contain the system model. So, the transfer function model mentioned in the previous comment should be added to figure 3.
· To perform a fair comparison, the authors should mention how they tuned the PID controller parameters. How can we know that these parameters are optimally chosen ?
· It is unclear why the mathematical model is extracted. Because, only experimental results are presented. So, to enrich the content, also simulation studies should be presented based on the mathematical model of the controlled system.
· The simulation and the experimental results should be compared with enriched case studies.
· The discussion of the results is poor. The authors should improve this section. This comment is also related with the previous comment. Both simulation and the experimental results should be presented and compared.
· The resolution of the figures should be improved and they must be centered in the pages.
· The vertical axis unit of figures 6 and 7 are not clear.
· To compare different controller performances, we are using different qualitative performance measures, such as integral square error (ISE), mean square error (MSE), and etc. These numerical results can be added in a table for each experimental study and the results can be discussed.
Author Response
Dear Reviewer:
Thank you for reading our manuscript “sensors-2086324”.We are very grateful for the editor and the reviewer’s valuable comments which have helped a great deal in revising the original manuscript. Our reply with detailed explanations is given below in the PDF file for the editor and the reviewer’s perusal.

Reviewer 2 Report
This paper explained the active disturbance rejection control (ADRC) optimized by Smith predictor (SP) is adopted to achieve constant tension control. The paper is well written and the analyses along with the results are well expressed. These are some of the comments that need to be addressed.
1. Abstract: should highlight the main idea of the method
2. Introduction: needs a stronger motivation.
3. Related work: missing, focus on recent works.
4. Quality of figures and tables are rather poor; some drawings are stretched; some are too small (Fig. 6, 7 for example).
5. Explain s SP-ADRC control algorithm ?
6. Clarify the difference between Tracking response and control error
7. What is the reason behind to compare with PID control? Why not consider recent papers?
8. Clarification is required in Eq. 13, references are missing for equations.
9. There is no comparison with recent controllers?
Author Response

(The authors gave the same response as above.)

Round 2
Reviewer 1 Report
Based on the corrections made, I think the manuscript can now be accepted after a minor English check.
Thank you..